# A Map Is a Living Structure with the Recurring Notion of Far More Smalls than Larges

**Bin Jiang** [1,*]  **and Terry Slocum** [2]

1    Faculty of Engineering and Sustainable Development, Division of GIScience, University of Gävle,
     SE-801 76 Gävle, Sweden
2    Department of Geography and Atmospheric Science, University of Kansas, Lawrence, KS 66045, USA;
     t-slocum@ku.edu
*    Correspondence: bin.jiang@hig.se

**Abstract:** The Earth's surface or any territory is a coherent whole or subwhole, in which the notion of "far more small things than large ones" recurs at different levels of scale ranging from the smallest of a couple of meters to the largest of the Earth's surface or that of the territory. The coherent whole has the underlying character called wholeness or living structure, which is a physical phenomenon pervasively existing in our environment and can be defined mathematically under the new third view of space conceived and advocated by Christopher Alexander: space is neither lifeless nor neutral, but a living structure capable of being more alive or less alive. This paper argues that both the map and the territory are a living structure, and that it is the inherent hierarchy of "far more smalls than larges" that constitutes the foundation of maps and mapping. It is the underlying living structure of geographic space or geographic features that makes maps or mapping possible, i.e., larges to be retained, while smalls to be omitted in a recursive manner (Note: larges and smalls should be understood broadly and wisely, in terms of not only sizes, but also topological connectivity and semantic meaning). Thus, map making is largely an objective undertaking governed by the underlying living structure, and maps portray the truth of the living structure. Based on the notion of living structure, a map can be considered to be an iterative system, which means that the map is the map of the map of the map, and so on endlessly. The word endlessly means continuous map scales between two discrete ones, just as there are endless real numbers between 1 and 2. The iterated map system implies that each of the subsequent small-scale maps is a subset of the single large-scale map, not a simple subset but with various constraints to make all geographic features topologically correct.

**Keywords:** wholeness; Christopher Alexander; third view of space; head/tail breaks; data classification; map generalization

---

*[Maps] are no longer merely considered as aids . . . , but as products of scientific research which, being complete in themselves, convey their message by means of their own signs and symbols and through these furnish the basis for further geographic deduction. . . . [the subjectivity] must not predominate: the dictates of science will prevent any erratic flight of the imagination and impact to the map a fundamentally objective character in spite of all subjective impulses.*

<div align="right">

Max Eckert (1908)

</div>

## 1. Introduction

The Polish mathematician Alfred Korzybski (1933) [1] first introduced the mantra *"a map is not the territory"* which points out two important facts about maps: a map has a similar structure to the territory, and a map is the map of the map of the map, and so on endlessly. This similar structure is

actually a living structure (Alexander 2002) [2] that possesses—in a recursive manner—far more small things than large ones. For example, a green tree with leaves is a living structure, because it has far more small branches than large ones, and importantly small branches are embedded in large ones. The notion of "far more smalls than larges" differs fundamentally from that of "more smalls than larges", as "far" indicates the distinct disproportionality between smalls and larges. This disproportionality is what underlies the 80/20 Rule or Pareto Principle (Koch 1998) [3]. The second fact is essentially derived from the first one, i.e., a map—due to the living structure of the territory—is considered to be the map of the map of the map, and so on endlessly. This is a recursive perspective through which all small-scale maps are subsets of the single large-scale map, and they all retain the underlying living structure. Motivated by the mantra *"a map is not the territory"* or more specifically by the two facts about maps, this paper argues that not only a territory but also its associated maps are a living structure, and it is the living structure of the Earth's surface or of a territory that makes maps and mapping possible.

Living structure is essentially a recursive and holistic view of looking at space or things in our environment. A living structure consists of only one type of recursively defined entities called centers, so the centers are made of the centers of the centers and so on, with far more small centers than large ones. For a street network, the individual junctions or street segments are not centers, but individual natural streets are centers; for a cartographic curve, its segments are not centers, but its bends are centers (see Figure 1 for illustration). Thus, the street network is a non-living structure when seen from the perspective of street junctions and segments, but it is a living structure when seen from the perspective of natural streets. In the same vein, the cartographic curve is non-living when seen from the perspective of individual line segments, but it is living when seen from the perspective of recursively defined bends. This living structure view of space is not typical of either Newtonian absolute space (the first view of space), Leibnizian relational space (the second view of space) or quantum mechanical space (Alexander 2002) [2]. Under the new third view of space conceived by Whitehead (1938) [4] and further developed by Alexander (2002) [2] space is considered to be neither lifeless nor neutral, but a living structure capable of being more alive or less alive. For example, a tree is more alive than its branches, and large branches are more alive than small branches. Importantly, the aliveness of a space is not determined by the space itself, but by those smaller spaces contained in it and the larger space that contains the particular space. In other words, the aliveness of space cannot be understood as a property of the space itself, or merely in terms of its own structure or shape. This is the essence of living structure.

It should be noted that the aliveness of a space has an iterative or accumulative property; this is essentially equivalent to the aphorism: *"the rich get richer and the poor get poorer"*. This paper is intended to reach a wide audience of both academics and laypersons, and thus we will not get into the iterative or accumulative nature of life (as shown in Figure 2), but rather focus on how a space is conceived as a coherent whole consisting of far more smalls than larges. The Earth's surface as a living structure is composed of far more ocean than land or far more small countries than large ones; a country is composed of far more small settlements than large ones (Christaller 1933, Zipf 1949) [5,6]; a settlement is composed of far more short (or less-connected) streets than long (or well-connected) ones; and a street is composed of far more small bends than large ones. It is this recursively defined space and importantly its inherent hierarchy of far more smalls than larges that makes maps and mapping possible.

This paper is further motivated by some fundamental questions on maps and mapping: What is the nature of maps? How do maps work? What does the image of the map look like? The third question is inspired by Lynch (1965) [7] and refers to the kind of mental image after one is exposed to a map rather than an actual city. This paper argues that the current state of the art of mapping practice or geographic representation is (mis-)guided by focusing largely on the notion of more or less similar things, although cartographers are guided—subconsciously or unconsciously—by living structure or its inherent hierarchy. For example, both raster and vector representations are not based on living centers; instead these representations are essentially nonliving or *"cold and dry"*, a term often used by Mandelbrot (1983) [8] to refer to Euclidean geometric shapes such as circles, rectangles, and straight

lines. In a vector representation, geometric primitives such as points, lines, and polygons are fairly good for computing processes and storages, but they are mechanistically imposed and are treated as fragmented pieces rather than living centers that can be well perceived as meaningful entities by human beings. Geographic features such as mountain ranges, river basins, settlements, street networks, and coastlines are actually meaningful entities, full of far more smalls than larges, so maps should be considered to be an iterative system, in which all subsequent small-scale maps can be automatically derived from a single large-scale map or database.

This paper is intended to establish living structure as a physical phenomenon and mathematical concept as a formal concept or foundation for maps and mapping. More specifically, the contributions of this paper can be highlighted as follows: Firstly, it is argued that not only the territory but also the maps are a living structure with the recurring notion of far more smalls than larges; second, it is demonstrated that the map is an iterative system, being the map of the map of the map, and so on endlessly; third, it is demonstrated how data classification and map generalization or mapping in general, can be considered to be a head/tail breaks process; finally, it is argued that objectivity should be favoured over subjectivity in maps and mapping, and maps are largely about the truth of the underlying living structure of the territory or the data.

The remainder of this paper is organized as follows. Section 2 examines the state of the art of maps and mapping in the context of geographic information science (GIScience) and how it is misguided by focusing largely on the notion of more or less similar things. Section 3 introduces the notion of living structure via four simple examples and its two fundamental laws. Section 4 demonstrates how maps, through data classification and map generalization, can be considered to be an iterative system. In Section 5, we further discuss the implications of living structure on maps and art. Finally, Section 6 summarizes our arguments and draws a conclusion.

## 2. The State of the Art of Maps and Mapping

Over a long history of mapping practices, cartographers have been guided by living structure, not explicitly or consciously but implicitly or subconsciously. That is why geographic features represented at different scales of the map look very similar not only to each other but also to the territory. This similarity is obvious across a wide range of map scales even when geographic features are represented by abstract symbols. For example, as map scale is reduced, city boundaries are represented in some gradually simplified manner, so the boundaries in small-scale maps are simplified versions of those in a large-scale map. As map scale is further reduced, all cities are collapsed into single dots, and only largest cities are retained in the smallest-scale maps. Despite the cartographers' instinct, living structure has not been well established as a formal concept in cartography and GIScience. Instead, in the current state of the art of maps and mapping or under the current mode of thinking, map making is constrained by a desire to portray geometric details of locations, sizes, and directions rather than the overall character of the territory being mapped. It was found that 85% people tend to see things sequentially or analytically by focusing on these details, while only 15% people see things figuratively or holistically (Alexander 2002, Alexander and Carey 1968, Alexander and Huggins 1964) [2,9,10]. Thus, there is little wonder that the current GIS representations focus on geometric details while miss the overall character. However, the figurative or holistic way is the right way to see the underlying character or living structure, which will be further discussed below.

Given their major concern with the locations, sizes, and directions of geographic features, current geographic information systems (GIS) represent these features by geometric primitives such as points, lines, and polygons (e.g., Clarke 1995, Kraak and Ormeling 1996, Chrisman 2001, Biang 2007, Longley et al., 2015) [11–15] Focusing on geometric primitives or geometric details tends to overlook the underlying living structure: the inherent hierarchy of "far more smalls than larges". In current GIS, a street network is considered to be a collection of more or less similar junctions or a set of more or less similar street segments between the junctions (Figure 1a). And a curvilinear feature on a map is considered to be composed of a set of more or less similar line segments (Figure 1c). Instead, a street

network should be considered as a set of far more less-connected streets than well-connected ones from the topological view of streets (Figure 1b). Equally, a curvilinear feature should be considered as a set of far more small bends than large ones (Figure 1d), recursively defined at different levels of scale. It should be noted that the non-living structure view (i.e., street segments and line segments) and the living structure view (i.e., streets and bends) are not contradictory but rather complementary to each other, with the former providing the geometric details, while the latter providing the overall character. As briefly mentioned above, the former view is under Newtonian and Leibnizian views of space (the first two views of space) being mechanistic (Descartes 1637, 1954) [16], while the latter view is under the third view of space—being organic (Whitehead 1938) [4]—developed by Alexander (2002) [2]: space is neither lifeless nor neutral, but a living structure capable of being more living or less living.

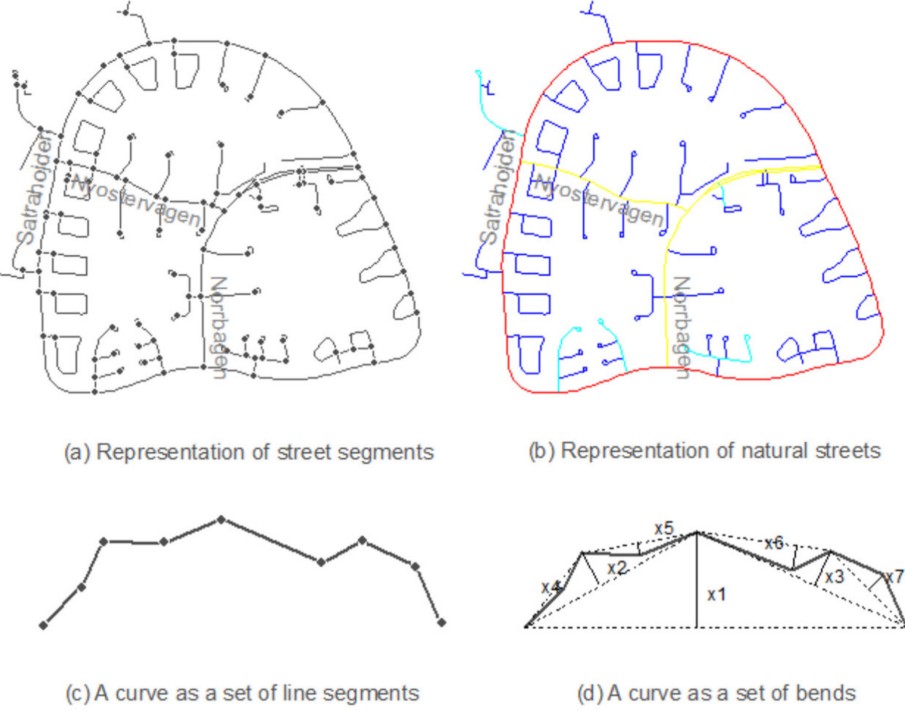

**Figure 1.** (Color online) Geometric primitives versus geometrically meaningful entities. (Note: A street network is represented as a set of junctions or street segments (geometric primitives, which are not centers) (**a**), whereas it is more correctly perceived as a collection of named streets (geometrically meaningful entities, which are centers) (**b**), each of which is colored as one of the four hierarchical levels: blue for the least connected streets, red for the most connected street (only one), and yellow and turquoise for those between the most and the least connected. A curvilinear feature is usually represented as a set of line segments (geometric primitives, which are not centers) (**c**), but it is more correctly perceived as a collection of far more small bends than large ones (geometrically meaningful entities, which are centers) (**d**), because the notion of far more small bends than large ones occurs twice: (1) $x_1 + x_2 + x_3 > x_4 + x_5 + x_6 + x_7$, and (2) $x_1 > x_2 + x_3$).

The perspective on geometric primitives is essentially a Euclidean geometric view, because it focuses on elements individually rather than on "far more smalls than larges" holistically, and on scales individually rather than on scaling collectively. The Euclidean geometric view enables us to see more or less similar things, e.g., more or less similar junctions (or equivalently more or less similar street segments) of a street network, or more or less similar line segments of a curvilinear feature (Figure 1a,c). This notion of more or less similar implies that there exists a characteristic mean for these things. More or less similar things are what underlies not only Gaussian statistics, but also the first law

of geography: *"everything is related to everything else, but near things are more related than distant things"* (Tobler 1970) [17]. The first law of geography or Tobler's law was formulated out of the notion of spatial dependency or autocorrelation. For example, your housing price is more or less similar to those of your neighbors, and today's weather is more or less similar to that of yesterday. Spatial dependency indicates a local fact, i.e., spatial events are not random, but autocorrelated. Tobler's law is what underlies many mapping activities, e.g., spatial interpolation for creating a smooth surface, and the kernel density estimation to create hotspot maps. Many geographic phenomena are indeed dependent on each other, so there are more or less similar things nearby or locally, but globally there are far more smalls than larges. For example, there are far more low-housing prices than high-housing prices. There are far more ordinary weather conditions than extraordinary weather conditions. This notion of far more smalls than larges has been formulated as the scaling law see (Section 3 for more details) (Jiang 2015) [18] (see Section 3 for more details). However, under the current mode of thinking with the assumption of more or less similar things, we have paid little attention to the notion of far more smalls than larges in maps and mapping. This situation is clearly reflected in data classification methods.

Natural breaks is commonly used to classify data for maps and mapping, and it is based on the principle that variation within classes should be minimized, while the variation between classes should be maximized (Jenks 1967) [19]. This principle is essentially based on the assumption that all classes have characteristic means. This is the same for the k-means clustering algorithm commonly referred to in computer science (Steinhaus 1956) [20]. This assumption does not hold for many real-world data, because they are heavy-tailed or long-tailed, like city sizes that follow a so-called rank-size distribution (Zipf 1949) [6]. For such data that have far more smalls than larges, it is wise to ask how many times the notion of far more smalls than larges recurs based on the data's inherent hierarchy. This is what motivated the head/tail breaks (Jiang 2013) [21] classification method for data with a heavy-tailed distribution, which has deep implications for living structure, as we will see in the next sections.

## 3. Living Structure of Centers and Its Two Fundamental Laws

All space and matter (either organic or inorganic) have some degree of life in it—*"every brick, every stone, every person, every physical structure of any kind at all"* (Alexander 2002) [2]—according to its underlying structure and arrangement. The phenomenon that all space has some degree of life has been extensively studied in computer science, architecture, and urban science (e.g., Gabriel 1998, Salingaros 2014, Jiang 2016, Mehaffy 2017, Gabriel and Quillien 2019) [22–26]. If the degree of life is too low, the structure is called a dead or nonliving structure; otherwise, it is called a living structure. Beyond the conventional notion of biological life as being self-producing (Schrödinger 1944) [27], the term life, also called wholeness, refers to the structural property of far more small centers than large ones. Here center refers to only one type of entity within a living structure, so centers are made of other centers recursively. There are far more small centers than large ones in any living structure in general. According to the definition of wholeness and centers, not only organic or alive things but also biologically dead things can have a living structure, as long as there is the recurring notion of far more small centers than large ones. For example, not only an alive tree, but also a dead tree is a living structure as long as the recurring notion of far more small branches than large ones remains. In one of his earlier works, Alexander (1979, p. ix) [28] referred to life (or wholeness) as a quality without a name, being *"the root criterion of life and spirit in a man, a town, a building, or a wilderness"*. Living structure pervasively exists in our environment, but the degree of living structure may not be so obvious. Let's look at some typical examples of living structure:

Example 1: A green tree with leaves.

A tree is made of far more small branches than large ones, out of which are made far more small branches than large ones, and so on. Thus, the tree is a living structure with the recurring notion of far more small branches than large ones. This is an alive tree, but its living structure remains even after it becomes dead, for the current notion of far more smalls than larges remains unchanged.

Example 2: The English country garden corner where a peach tree grows against a wall (Alexander 1979) [28]

> *The wall runs east to west; the peach tree grows flat against its southern side. The sun shines on the tree and as it warms the bricks behind the tree, the warm bricks themselves warm the peaches on the tree. It has a slightly dozy quality. The tree carefully tied to grow flat against the wall; warming the bricks; the peaches growing in the sun; the wild grass growing around the roots of the tree, in the angles where the earth and roots and wall all meet.*

Example 3: The Berkeley street corner at the intersection of Hearst and Euclid (Alexander 1965) [29]

> *In Berkeley at the corner of Hearst and Euclid, there is a drugstore, and outside the drugstore a traffic light. In the entrance to the drugstore there is a newsrack where the day's papers are displayed. When the light is red, people who are waiting to cross the street stand idly by the light; and since they have nothing to do, they look at the papers displayed on the newsrack which they can see from where they stand. Some of them just read the headlines, others actually buy a paper while they wait. This effect makes the newsrack and the traffic light interactive; the newsrack, the newspapers on it, the money going from people's pockets to the dime slot, the people who stop at the light and read papers, the traffic light, the electric impulses which make the lights change, and the sidewalk which the people stand on form a system - they all work together.*

Among the above three examples, the first one is the most obvious, while the other two may seem a bit obscure. In the first example, the tree is considered to be the living center, consisting of far more small centers (actually branches) than large ones, which further consist of far more small centers than large ones, and so on. The tree can be considered the center of the center of the center, and so on. The country garden corner is the first place where Alexander (1979) [28] was pondering on the phenomenon of life and struggling with its naming as the quality without a name, the precursor of living structure. The garden corner as a living center consists of many centers, among which the most salient include the peach tree, the wall that is composed of many bricks, the wild grasses, the earth and the sun. All these living centers, which are certainly living structures, constitute mutual supporting relationships, e.g., the light shines on the wall, warming the bricks and dead leaves, and grasses nourish the earth and the tree, forming a coherent whole or ecological system. The corner is a living center of the larger center (e.g., the garden), which is a center of an even larger center (e.g., the neighborhood) of an even larger center (e.g., the city), and so on endlessly towards the entire universe.

The notion of far more smalls than larges can be rephrased in some situations as that of far more less-used locations than well-used ones, or far more meaningless locations than meaningful ones. The street corner scene consists of a few well-used locations (the newsrack, the traffic light, and the pavement between them) and many remaining less-used locations that hold together as a coherent living structure. The well-used locations occupy a small amount of space but receive far more attention than the less-used locations; thus, there are far more smalls than larges in terms of how much space is used, or how meaningful semantically space is perceived. In other words, the small amount of space is well-used or more meaningful, whereas the large amount of space is less-used or less meaningful: this is truly a living structure. These well-used locations can be compared to the eyes, nose and mouth of a human face, which is surely a living structure; the eyes, the nose and the mouth occupy a small amount of space, but they receive a large amount of attention, whereas the remaining parts of the face occupy a large amount of space, but receive a small amount of attention as shown by eye-tracking experiments (Yarbus 1967) [30]. In any living structure, there is a recurring notion of far more low-intensity centers than large ones. To make this point clear, let's now look at a fourth example.

Example 4: A ten-city cluster as a living structure (Figure 2a)

A ten-city cluster within a square space in which there is one largest city (red, Figure 2b), two middle-sized cities (green, Figure 2c), and seven smallest cities (blue, Figure 2d). There are three hierarchical levels among the ten cities indicated by the three different colors. In other words, the notion of far more smalls than larges recurs twice, so it is a living structure. The nested and mutual relationship among the ten cities constitutes a complex network (Figure 2e); note that the relationship at the same

level is undirected, while the relationship between different levels is directed. The degree of life can be computed relying on Google's PageRank (Page and Brin 1998, Jiang 2015) [31,32], leading to far more low-life centers than high-life ones (Figure 2f). The PageRank way of computing the degree of life of a center can be compared to assessing how important a person is, for which we should ask how important not only his/her friends are, but also the friends of the friends of the friends, and so on virtually for all people on the planet. In this sense, note that the small northcentral city has a higher degree of life than other small cities because of the more links it receives and the few links it gives out. In the same vein, the right green city has a higher degree of life than that of the left one.

In considering these examples (in particular examples 2 and 3), the reader may be thinking that the living structure approach is subjective, as people may have different thoughts about these situations, but Alexander argued that the approach is largely objective and is not just a matter of personal opinion and taste (Alexander 1979 [28], Alexander 2002 [2]). This situation may be compared to the situation when asking a group of people about the temperature of a piece of ice and a glass of hot water; there is little doubt that the hot water has a higher degree of temperature than the ice, although people may disagree on the exact temperature of the hot water. Living structure is not only objective and precise, but also reflected in the human mind and heart. It was found through the mirror-of-the-self experiment that living structure correlates with very personal questions such as whether I feel myself whole, or whether my spirit is lifted up in the presence of living structure (Alexander 2002 [2], Wu 2015 [33]). Although there is indeed some personal opinion and human taste involved in interpreting living structure, the mirror-of-the-self experiment showed that living structure is a shared notion among a majority of people regardless of their faiths, cultures, and ethics.

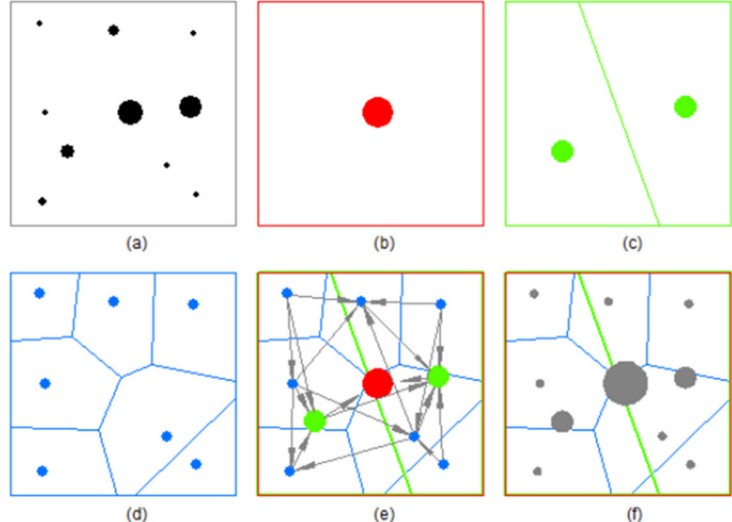

**Figure 2.** (Color online) The ten fictitious cities and their interrelationship constitute a living structure. (Note: As a structural invariant of the central place theory model (Christaller 1933) [5], the cluster of the ten cities (**a**) is composed of the largest city (**b**) bounded by the red square, surrounded by two middle-sized cities (**c**) separated by the green line and bounded by the green box, and further surrounded by seven smallest cities (**d**) separated by blue lines and bounded by the blue box, thus with three hierarchical levels, indicated by dot sizes and colors. Because of mutual relationship among the ten cities (**e**), each city has different degree of life, as indicated by the dot sizes (**f**)).

Underlying any living structure there are two governing laws: the scaling law across all scales (Jiang 2015) [18] and Tobler's law at each scale (Tobler 1970) [17] (Table 1). According to the scaling law, there are far more low-life centers than high-life ones across all scales ranging from the lowest to the highest. It is important to note that the scaling law is a relaxed version of other long-standing laws or rules such as Zipf's law (1949) [6] and the universal rule (Salingaros and West 1999) [34] which all require a power law distribution. Instead, the scaling law requires only that far more smalls

than larges recurs at least twice with the ht-index being at least three. From a dynamic point of view, the ht-index of a living structure usually will increase as time goes by, and when it reaches to 6 or 7, the data distribution may demonstrate a power law. While the scaling law applies to all scales, Tobler's law applies to each scale, i.e., centers tend to be more or less similar on each scale. These two laws characterize living structure from both global and local perspectives. However, the scaling law is primary and global, while Tobler's law is secondary and local. Any structure with more or less similar centers globally tends to be uninteresting or less-living. As we have shown in Figure 1, a street network if seen from the perspective of individual street segments – the geometric view – tends to be uninteresting or less-living, but it is a living structure if seen from the topological perspective of connected streets, and a curvilinear feature when seen from the perspective of segments tends to be uninteresting or less-living, but it is a living structure when seen from the perspective of the recursively defined bends.

**Table 1.** Two fundamental laws of living structure. (Note: These two laws—scaling law and Tobler's law—complement each other and recur at different levels of scale of living structure).

| Scaling Law | Tobler's Law |
| --- | --- |
| There are far more small centers than large ones | There are more or less similar centers |
| across all scales, and | available at each scale, and |
| the ratio of smalls to larges is disproportional (80/20). | the ratio of smalls to larges is closer to proportional (50/50). |
| Globally, there is no characteristic scale, so exhibiting | Locally, there is a characteristic scale, so exhibiting |
| Pareto distribution, or a heavy-tailed distribution, | Gauss-like distribution, |
| due to spatial heterogeneity or interdependence, indicating | due to spatial homogeneity or dependence, indicating |
| complex and non-equilibrium phenomena. | simple and equilibrium phenomena. |

It should be noted that the scaling law or scaling hierarchy is what underlies many natural and societal phenomena such as coastlines, terrain ranges, earth quakes, and financial markets. They are also called complex systems which demonstrate non-equilibrium character (e.g., Simon 1962 [35], Zipf 1949 [6], Mandelbrot 1983 [8], Bak 1996 [36]). Complex systems appear complex and non-equilibrium at the global scale, but they may demonstrate simple and equilibrium character at a local scale. Climate is essentially a complex system, so it is unpredictable essentially; there are far more ordinary weather conditions than extraordinary ones globally and in a large time scale, but locally and in a small time scale, today's weather is more or less similar to that of yesterday. It should be also noted that the notion of far more smalls than larges relies on its recurring nature. In other words, the notion must recur at least twice rather than just once. There are three different perspectives to assess whether there are far more smalls than larges: topological, geometrical, and semantic. Among the three perspectives, the topological is primary because it specifies the spatial configuration or the underlying structure that determines the degree of life. For example, a tiny city in the middle of a set of cities may look extremely small, but it tends to have a high degree of life because of its many connections. Herewith it is not geometric size, but the topological connection that essentially determines the degree of life, as we can see for the small northcentral city in Figure 2.

## 4. Mapping as the Head/Tail Breaks Process: Data Classification and Map Generalization

We contend in this paper that both maps of different scales and the territory are living structures with the inherent hierarchy of far more smalls than larges. Map making essentially depicts the inherent hierarchy or the recurrent notion of far more smalls than larges. Let us now illustrate how data classification and map generalization, or mapping in general, can be conducted as a head/tail breaks process relying on two simple examples.

Let's take first look at a set of 39 data values: (1, 1/2, 1/3, 1/4, ... , 1/39), which follow exactly a rank-size distribution (Zipf 1949) [6] which means the first largest city is twice as big as the second largest city, three times as big as the third largest city, and so on. This dataset has four inherent

hierarchical levels, for the notion of far more smalls than larges recurs three times as shown in Table 2. More specifically, the average of the 39 values is ~0.11, which partitions these values into two groups: those greater than the average is called the head: (1, 1/2, 1/3, . . . , 1/9), and those less than the average is called the tail: (1/10, . . . , 1/39). In this case, we can see in Table 2 that the head has nine large values and the tail has far more small values (thirty). Among the nine in the head, their average is ~0.31, which further partitions these nine into two groups: three large in the head: (1, 1/2, and 1/3), and six small in the tail (1/4, 1/5, . . . , 1/9). For the three in the head, their average is ~0.61, which further partitions the three into two groups: one large (1) for the head, and two small (1/2, 1/3) for the tail. This iterated head/tail breaks process relies on the data itself to partition the data; thus, the resulting classes or hierarchical levels are determined from the bottom up rather than imposed from the top down or by cartographers. In other words, head/tail breaks lets the data determine classes, in terms of not only how many classes, but also how to set class intervals. Real world datasets are often much larger or more complex than this simple dataset, but the underlying data classification approach remains the same in order to uncover the inherent hierarchy of data for mapping. Figure 3 illustrates the head/tail breaks classification for the dataset with four inherent hierarchies.

**Table 2.** Statistics of the head/tail breaks process of the 39 numbers.

| Number | Mean | # Head | # Tail | % Head | % Tail |
|---|---|---|---|---|---|
| 39 | 0.11 | 9 | 30 | 23% | 77% |
| 9 | 0.31 | 3 | 6 | 33% | 67% |
| 3 | 0.61 | 1 | 2 | 33% | 67% |

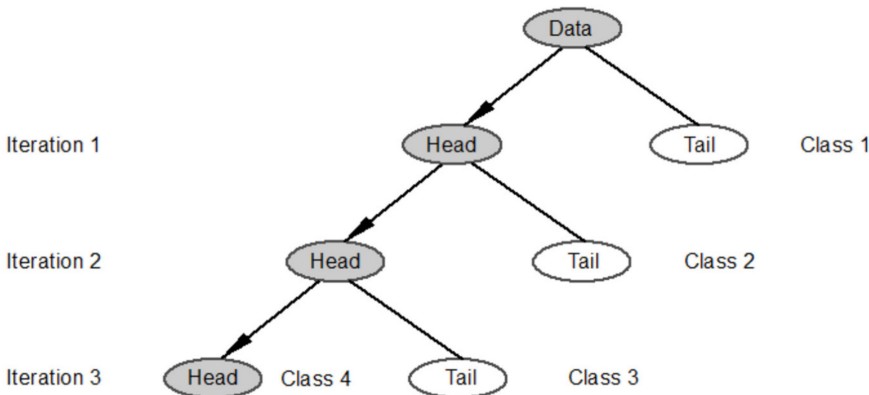

**Figure 3.** Illustration of the head/tail breaks as an iterative function. (Note: The data as a whole is recursively divided into the head (for those greater than the average) and the tail (for those less than the average). The whole or data is seen as an iterated system, i.e., the head of the head of the head and so on. For the sake of simplicity, we illustrate three iterations or four classes).

The reader may wonder at what time or under what condition, the iteration process should stop. This is a valid question. Roughly put, head/tail breaks has to ensure, at each iteration, that the head percentage is far smaller than that of the tail, to reflect the desire for "far more smalls than larges". One reasonable solution based on the first author's experience is that the number in the head should be less than 40%. However, with many real-world data, the first iterations may end up with more than 40% for the head, but for all subsequent iterations the head percentages are all around 20%. Therefore, there are two different ending conditions for head/tail breaks: (1) the head percentage must be less than 40% for every iteration, and (2) the average head percentage for all iterations must be less than 40%. Given the complexity of the real-world data, the second is more preferred, for it is less restrictive than the first.

Slocum et al., (2008) [37] discuss numerous criteria for determining an appropriate method of classification. From the above computations, we can see that head/tail breaks does an excellent

job of meeting two key criteria. First, by iteratively assigning "far more smalls" at each step of the classification, head/tail breaks pays careful attention to the graphical distribution of the data along the number line (e.g., expressed as either a dot plot or a rank-size plot). Second, the method provides an objective approach for stopping the classification process. The latter is in contrast to the natural breaks method, which generally determines the number of classes through a visual examination of a graph of the number of classes plotted against a goodness-of-variance fit measure.

Having discussed data classification, let us showcase how map generalization can be conducted through head/tail breaks. We will use the Koch curve (Figure 4a, Koch 1904 [38]) as a working example. Let's first consider how the Koch curve can be generated as an iterative function. A segment of unit 1 is divided equally into thirds, with the middle third replaced by two sides of an equilateral triangle. This process can be repeated until the scale is infinitely small. To illustrate, we show the first four iterations in Figure 4a and Table 3. This is one of the first classic fractals based on a rigorous definition. The reader may argue that cartographic objects do not exhibit the regularity of the Koch curve, as cartographic objects have a much more random character. However, the notion of far more smalls than larges remains for both the classic and randomized Koch curve. For this reason, and for the sake of simplicity, we use the classic Koch curve for illustration of our ideas in this paper. The Koch curve is a deterministic process of creating fractals: as the scale drops by 1/3, the number of segments increases by 4 times, and the direction of the little bump is always in the same direction. When this deterministic process changes to a non-deterministic one, the Koch curves would look like coastlines, clouds, or city skylines. However, the notion of far more smalls than larges would remain unchanged.

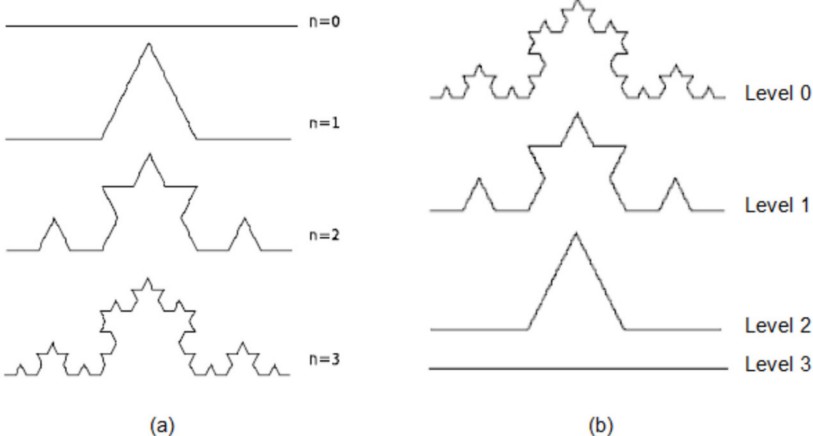

(a)  (b)

**Figure 4.** Generation (**a**) and generalization (**b**) of the Koch curve with the first four iterations (Note: Beginning with a segment of scale 1 (n = 0), it is divided into thirds, and the middle third is replaced by two equilaterals of a triangle, leading to four segments of scale 1/3 (n = 1). This division and replacement process continues for scales 1/9, and 1/27, leading respectively to 16 segments, and 64 segments (n = 2, and 3). This is the generation of the Koch curve, as shown in panel (**a**). On the other hand, the Koch curve (Level 0) can be generalized in a step-by-step fashion, as shown in Table 3, resulting in the outcome in panel (**b**)).

**Table 3.** The four iterations of the Koch curve as shown in Figure 4a.

| Iteration | Scale | # Segment |
|:---:|:---:|:---:|
| 0 | 1 | 1 |
| 1 | 1/3 | 4 |
| 2 | 1/9 | 16 |
| 3 | 1/27 | 64 |

Let's take the Koch curve at the fourth iteration (n = 3 in Figure 4a or Level 0 in Figure 4b) to see how it can be generalized or simplified at the different levels of detail. It is very important to realize

that the Koch curve is not just a collection of 64 segments with the size of 1/27 (or ~0.04). It is a wrong way of thinking based on Euclidean geometry. The right way of thinking—or being recursive or the way of living structure—is that it is a collection of 64 segments of size 1/27 (or ~0.04), plus 16 segments of size 1/9 (or ~0.11), plus 4 segments of size 1/3 (or ~0.33), plus 1 segment of size 1, i.e., 64 + 16 + 4 + 1 = 85 segments (Table 4). In other words, all large sizes (1, 1/3, and 1/9) are embedded in the small one (1/27) in a recursive manner. Now let us calculate the average length of these 85 segments, which is ~0.08. Clearly, there are 16 + 4 + 1 = 21 long segments (longer than the average) and 64 short segments (shorter than the average) (Table 4), implying far more short segments than long ones. For the purpose of generalization, we retain the long ones, so we obtain the result of level 1 in Figure 4b. For level 1, there are 21 segments, and their average length is ~0.20 (Table 4). There are five segments greater than this average ~0.20, and thus we obtain the level 2 generalization (Figure 4b). For level 2, there are 5 segments, and their average length is ~0.47. There is only one segment greater than this average ~0.47 (Table 4), and the result is level 3 (Figure 4b). From this generalization process, we can remark that the curve is the bump of the bump, the bump of the bump, and the bump of the bump (actually the last iteration is no longer a bump, but a straight line).

**Table 4.** Statistics of the head/tail breaks process for the Koch curve.

| # Segment | Mean | # Head | # Tail | % Head | % Tail |
|---|---|---|---|---|---|
| 85 | 0.08 | 21 | 64 | 25% | 75% |
| 21 | 0.20 | 5 | 16 | 24% | 76% |
| 5 | 0.47 | 1 | 4 | 20% | 80% |

Seen from the working example of the Koch curve, map generalization is no more than retaining large things, while eliminating small things in a recursive manner to get different levels of detail. Geographic features like coastlines may look much more complicated than the Koch curve, but the head/tail breaks principle remains the same. A cartographic curve may be represented as a set of bends, recursively defined by three points as illustrated in Figure 1d. The largest bend $x_1$ is followed by the two middle-sized bends $x_2$ and $x_3$, and the four smallest bends $x_4$, $x_5$, $x_6$, and $x_7$. From the point of view of the recursively defined bends, there are three inherent hierarchical levels. Thus, the simplification of the line can be carried out in a similar manner as that of the Koch curve. However, there is one potential problem in the course of line simplification or generalization: The simplified curve may create intersections either with the curve itself or with other geographic features, the so-called "self-intersection" or "intersection with others", which produces topologically incorrect geographic features. The solution to this problem is simple (Jiang 2017 [39] and related references therein) whenever intersections occur with a simplified curve, that part of the curve has to go back to the previous iteration, or a few trivial points have to be added to avoid the intersection, but all other parts without conflicts remain unchanged.

Seen from these two examples, data classification and map generalization can be accomplished objectively, by applying head/tail breaks to the underling living structure. Since the resulting maps are automatically determined by head/tail breaks, we can say that *"the data speak for itself"*. Under the current mapping paradigm, there are many parameters to be set carefully to fulfill automatic map generalization between two particular map scales (e.g., from 1:10 K to 1:50 K) (e.g., Stoter et al., 2014, Buttenfield et al., 2011) [40,41]. In contrast, under the living structure view, and relying on the head/tail breaks, there are few parameters to be set; if any parameters are set, the idea is to let the data determine a meaningful cutoff rather than have cartographers to make this decision. This opens up the possibility of automatic map generalization from a single large-scale database to create a large variety of small-scale maps.

## 5. Further Discussions on Living Structure for Maps and Art

Maps are essentially about the truth of the territory or that of the Earth's surface, or more specifically the truth of the underlying living structure. There is little doubt that there are subjective

factors involved in maps and mapping (Monmonier 1991, Wright 1942) [42,43], but maps are largely about truth, and an important component of this truth is the inherent hierarchy of far more smalls than larges. Objectivity is more important than subjectivity in the mapping process rather than the other way around. It is for sure that certain aspects are sometimes distorted or exaggerated. For example, for the purpose of navigation, the London Underground Map puts linkages and stops in the first and foremost priority of representation, while drastically simplifying locations, routes, and even orientations. The Underground Map, although drastically simplified, becomes far more informative for the purpose of navigation. As another example, Figure 5 shows the topology of underlying streets, indicating the living structure of far more less-connected streets than well-connected ones. Note that the graph in Figure 5b is not georeferenced at all, neither the nodes nor the links have any georeferenced information. Instead, the node sizes indicate the degree of connectivity of the corresponding streets in Figure 5a. With the graph, the underlying living structure of the street network becomes very striking. It helps to answer such a question as: how many intermediate streets does one have to pass through in order to go from location A to location B? Unfortunately, this question is virtually impossible to address with any conventional representation (e.g., Figure 1a). In this case Figure 5b provides a good example about the true living structure of the Earth's surface.

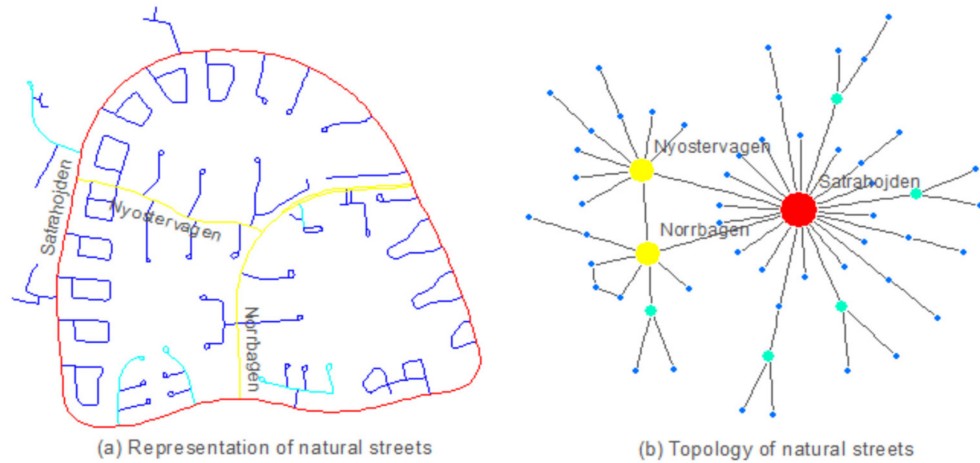

**Figure 5.** (Color online) A living structure with four hierarchical levels of natural streets. (Note: The natural streets that are represented—on the surface—by geometrical details of locations, sizes, and directions (**a**) are transformed into the topology of the streets or living structure, in the deep sense, with far more less-connected streets than well-connected ones (**b**).

The living structure view of maps and the territory or the Earth's surface in general is a powerful concept that has important implications for maps and mapping. The implications touch such issues as automatic mapping, the nature of maps, the mental image of maps, and how maps work. Although a large body of research has attempted to address these profound issues in the past century (e.g., Eckert 1908, Robinson 1952, MacEachren 1995, Wuppuluri and Doria 2018, Tversky 2019) [44–48], such research has not considered the underlying living structure. It can be argued that the nature of maps depends to a considerable extent on the underlying living structure or the inherent hierarchy of far more smalls than larges; it is through the living structure or the inherent hierarchy that maps often convey useful information, and those largest and most connected and most meaningful things constitute the mental image of the map in the human mind. In this sense maps, in particular terrain maps or topographic maps, are not very different from products of art (like landscape paintings), because both tend to reflect the underlying living structure.

In addition to revealing the living structure of far more smalls than larges, another advantage of Alexander's approach is that the resulting maps should provide a sense of good feelings such as belonging, healing, and well-being. This concept may seem foreign to cartographers, but it is integral to the concept of living structure. For example, Alexander (2002) [2] describes the good feelings that

come from viewing architecture that is based on living structure principles. We conjecture that it is the underlying living structure of natural scenes that has the healing effect on patients' recovery from surgery (Ulrich 1984) [49]. In this sense, it also can be argued that there is little difference between when people are exposed to good maps and when people are exposed to good scenes (either landscapes or indigenous buildings). In other words, maps can be viewed as no different from landscapes and buildings, because they are all living structure. Also, in this connection, there is little difference between maps and fine products of art, because both capture the underlying living structure. Actually, not only fine products of art, but also abstract paintings reflect the underlying living structure. For example, Jackson Pollock's epic painting Blue Poles: Number 11, 1952 presents nothing more than the recurring notion of far more smalls than larges, which is the very essence of nature or of the Earth's surface in particular. The fractal or living nature of Jackson Pollock's drip paintings was verified by the physicist Taylor (2006) [50], who discovered that those paintings with a fractal dimension around 1.3–1.5 tend to have the highest aesthetic appeal or the highest degree of beauty in the human mind or heart.

As another example of the relevance of art, Piet Mondrian's compositions of red, yellow, and blue are living structures, although they are far less living than those of Pollock's drip paintings. Mondrian's compositions present nothing more than the recurring notion of far more smalls than larges. Figure 6 presents one of Mondrian's compositions, consisting of only seven simple pieces. It does not exhibit extensive living structure, but we see clearly twice the recurring notion of far more smalls than larges. The composition painting can be viewed as the product of the differentiation process in a step-by-step fashion, i.e., the square space is continuously differentiated leading to far more smalls than larges. Viewed the other way around, the empty square can be viewed as the outcome of aggregating or clustering small pieces into large ones, also in a step-by-step process. In spirit, the aggregating process is actually the map generalization process, as small pieces are aggregated to large ones.

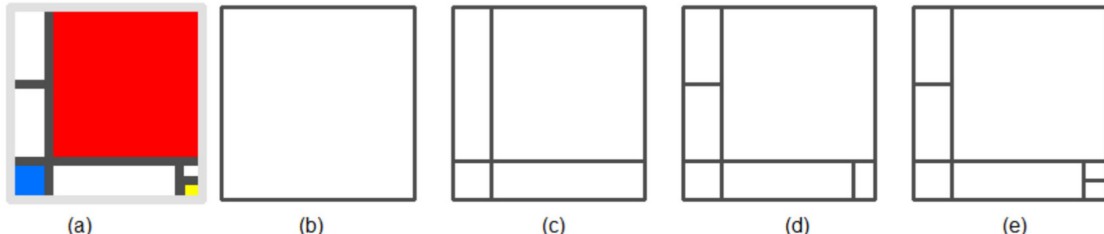

**Figure 6.** Composition II by Piet Mondrian (**a**) and its evolution from the empty square. (Note: It meets the minimum condition of being a living structure, and it is simple enough to illustrate how it is differentiated in a step-by-step fashion in panels (**b**–**e**), thus with a gradually increasing degree of life or beauty; there are far more smalls (4) than larges (1) from (**b**,**c**), and again far more smalls (6) than larges (4) from (**c**,**d**), so the ht-index is 3, which meets the condition of being a living structure. Thus both (**b**,**c**) are non-living structure, for their ht-index is less than 3. In addition, there is a violation of far more smalls (7) than larges (6) from (**d**,**e**), for 6 and 7 are more or less similar. If we consider the evolution in the opposite direction (from (**e**,**b**) then it can be viewed as a generalization process, very much like that of map generalization).

## 6. Conclusions

Beginning with the two basic facts about maps, we have attempted to argue that not only maps but also the territory is a living structure. The living structure exhibits the inherent hierarchy of far more smalls than larges, e.g., far more low peaks than high peaks over a terrain surface, far more small cities than large ones in a country, far more short streets than long ones in a city, and far more small bends than large ones over a coastline. Although cartographers have long been, subconsciously or unconsciously, guided by living structure for map making or map reading, this paper is intended to explicitly establish living structure as a formal concept of maps and mapping. We have argued and demonstrated that it is the inherent hierarchy or the recurring notion of far more smalls than larges

that makes maps and mapping possible, and data classification and map generalization within map making can be accomplished through the head/tail breaks process. Viewed in the reverse sense, it is essentially the conventional mode of thinking based on Euclidean geometry and Gaussian statistics that makes automatic map generalization virtually impossible.

Maps can, should, and must be treated as a scientific product, and their quality can be judged to a considerable extent by the underlying living structure. It can be argued that the more that maps reflect the underlying living structure, the better the quality of the maps. The iterative head/tail breaks method provides an objective method of determining an appropriate data classification or level of generalization, and can avoid some of the subjective issues associated with more traditional approaches. A map based on living structure can evoke a good sense of feeling in human beings in their deep psyche in terms of belonging, healing, and well-being. This kind of good feeling can explain why many people love maps. Importantly, this kind of feeling is shared. To paraphrase Alexander (2002) [2], a majority of our feelings are shared, and idiosyncratic feelings account for only a minority. Using Alexander's approach, objectivity is favored over subjectivity in maps and mapping, because maps and the territory share the same living structure. In this new way of mapping, there is a distinct attempt to design from the bottom up, i.e., let the geospatial data map or speak for itself. This new way of mapping departs radically from the current ways of mapping often involving many parameters imposed by techniques (e.g., natural breaks, Kernel density function) or by cartographers.

**Author Contributions:** Writing—original draft, Bin Jiang; Writing—review & editing, Bin Jiang and Terry Slocum. All authors have read and agreed to the published version of the manuscript.

**Funding:** The paper was partially supported by the Swedish Research Council FORMAS through the ALEXANDER project with grant number FR-2017/0009.

**Acknowledgments:** This paper grows out of collaboration between the two authors on an AutoCarto 2020 Workshop on Living Structure as a Scientific Foundation of Maps and Mapping. The workshop has been delayed due to the COVID-19 pandemic. During the preparation of the workshop, the two authors have exchanged numerous emails discussing the two tools Axwoman and head/tail breaks. One of the fruits of the collaboration is this paper, arguing for establishing living structure as a formal concept or foundation of maps and mapping.

**Conflicts of Interest:** The authors declare no conflicts of interest.

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
