# Peer review of "A Map Is a Living Structure with the Recurring Notion of Far More Smalls than Larges"

_ijgi, doi:10.3390/ijgi9060388_

Round 1

Reviewer 1 Report

This is an interesting and innovative study in geography. It goes to the very heart of cartography, questioning some recent approaches to the discipline. Aside from one basic disagreement on the approach and terminology, I find its results both interesting and relevant. 

The authors use the work of Christopher Alexander, an architect-mathematician who has derived scaling laws of structure. Alexander does establish the relationship between certain mathematical structures and living forms, but not the present paper. It is not enough to refer to Alexander, because here there is a specific set of problems not directly addressed by Alexander. The authors have to do additional work to prove their point and to indicate what is and what is not living structure. 

What I don't like in this paper is the trivialization of fractal scaling to the simplistic formula “far more smalls than larges”. Even when discussing an exact fractal like the Koch curve, which has a precise inverse-power law distribution, the authors go back to their oversimplified term. I believe this insistence diminishes the paper for no reason, and even confuses those who are already familiar with fractal scaling. 

The "scaling law" used here is just a Pareto distribution, which vastly oversimplifies the fractal distribution. Measuring the length of a coastline is the classic origin of fractals, but again, there is an attempt to reinvent everything and to avoid referring to fractal scaling. 

A key reference is missing: 

Salingaros, N. A. and West, B. J. (1999) A universal rule for the distribution of sizes, Environment and Planning B: Planning and Design, 26, 909-923.

which covers the Koch curve exactly, and describes fractal scaling accurately. 

Some technical problems: on pp. 4, 11 the subscripts are not well indicated in the text. 

The authors criticize other authors, that "such research has not considered the underlying living structure". I agree. But Pareto's law is not enough to define living structure. 

In conclusion, the examples provided are nice to show the relationship with the fine arts, but they are only distantly related and not conclusive. The authors' argument would be boosted by examples from geography instead. 

Author Response

See details in the attached file

Reviewer 2 Report

It is an interesting theoretic paper with new ideas and propositions in the field of geospatial analysis and mapping. The paper continues the dialogue on previously introduced novel ideas and methods.

However, the manuscript has mainly an explanatory character, which develops quite well, whereas the propositions remain weak, or anyway, not well understood.

Most sections rely exclusively on examples, whereas logical arguments are missing or are not well justified. In addition to that, the different sections of the manuscript are not well linked between them - they look like independent chapters of a book rather than paper sections.

It is true that most of the sections achieve their explanatory role, but they do not justify their role in reaching the overall objective of the paper (or the ‘wholeness’ is lost). In other words, it is not made clear how exactly fractals link with head/tail breaks, and in turn, how the latter can express or serve hierarchical structures (or living structures) and complexity.

Methodologically speaking, use of head/tail breaks paradigm seems to be the central idea and proposition of the paper for data classification and mapping generalization. However, it is questionable if this is enough to account for the local heterogeneity, as it relies on global statistics only; and local heterogeneity is a key issue in geospatial analysis and mapping.

In terms of development, the manuscript is lacking clear definitions, even for the main concepts, such as the ‘living structure’; the latter is referred 83 times (!) in the manuscript, but a true definition cannot be found (only imposed, but still vague). Could ‘living structure’ possibly be considered as the hierarchical structure underlying in a set? Also, some concepts are vague or not well-established, e.g. 'degree of life' (which is understood as importance of a feature in a set, or feature’s position in the hierarchy).

Finally, many arguments are repeated several times, even in the same paragraph, without conveying new information. An idea would be to use terms taken from the Set Theory and GIS, to avoid vagueness and misunderstanding. In other words, it seems that terminology used by Alexander dominates over most common though well-established terms and definitions.

It is suggested that the manuscript is rewritten with clear objectives and results, new abstract (avoid very long quoted text), certainly new introduction (clear objectives), and sections linked together and contributing to the overall concept in an understandable way.

The geospatial context is not always apparent and true mapping examples are missing. An experiment on true geographic data (rather than abstract spatial models) might be necessary for better demonstration. 

Author Response

See details in the attached file

Round 2

Reviewer 1 Report

I'm disappointed in the revision of this manuscript. My original concerns are now addressed in a tacked-on Note 1 at the end of the paper. This is not an in-depth discussion of fractal scaling as requested. The paper retains its original reduced model of a distribution that appeared to me as a trivialization of fractal scaling. Surely the authors can do better than this. 

Also, what is added is the term "copy of a copy". I won't judge whether this helps to boost the argument. But I would recommend establishing an analogy of information loss when this process occurs in reproducing images or music files -- they lose accuracy. Just an optional suggestion to the authors. 

I'll be very happy to approve this paper when my original concerns are addressed, and not brushed off in a footnote.

Author Response

Reviewer 2:

Again, many thanks the second reviewer for his/her patient discussion. In this round, we made some major revisions:

(1) a formal definition of centers, just one sentence just before the four examples. “Here center refers to only one type of entity within a living structure, so centers are made of other centers recursively.” This definition sounds very abstract, and it is somehow like what is object in object-oriented programming, everything is an object. Given the abstract nature of the definition, the four examples make a good sense to further illustrate what center is.

(2) We added a remark about 85% of people see things analytically, while only 15% of people see things figuratively or holistically. “It was found that 85% people tend to see things sequentially or analytically by focusing on these details, while only 15% people see things figuratively or holistically (Alexander 2002–2005). Thus, there is little wonder that the current GIS representations focus on geometric details while miss the overall character. However, the figurative or holistic way is the right way to see the underlying character or living structure, which will be further discussed below.”

(3) Figure 2 is substantially revised from previously two panels to now six panels.

I have already raised my considerations, in which the authors have replied -although not always convincingly.

Sorry, we are not convinced by your statement. Could you be specific on which points are not convincing? So that we could have some further discussions. This is really what we have been advocating on living structure as a formal concept of maps and mapping. This may sound like our opinion, but the opinion is well supported by evidence, e.g., Jiang 2017 and the related references therein. This paper demonstrates the evidence via two examples in Section 4. 

I propose acceptance after minor corrections provided it is an opinion paper, which brings some fresh air in thinking about maps, but not a research nor a review paper.

As we argued previously, this is a paradigm-oriented paper rather than ordinary technical-oriented paper. However, all argued points are supported by evidence and demonstrated by examples. That we chose not to cite some of the previous papers is to make the paper as much self-contained as possible and try our best to avoid too many self-references.

The link http://lifegis.hig.se/Sweden/ is missing some guide; also, it is hard to understand the differences from a -say- Google Map. And what is the difference between the two sides?

As the reviewer has sensed that here is little difference between the two sides. The little difference indicates that human cartographers have been – subconsciously or unconsciously – guided by living structure. However, living structure has not yet been considered to a formal concept in cartography or in maps and mapping practice. This is exactly what this paper intends to do.

Still, the green diagonal line is not explained in Figure 2; please, write that the green line divides space into two sections, as the blue lines divide space into seven sections, etc.

Sorry, we have made this correction and substantially revised the figure from the previous two panels to now six panels as mentioned above. Many thanks!

Although in the title you claim that 'A Map Is a Living Structure ...', in the Conclusions you claim '...we have attempted to argue that not only maps but also the territory is a living structure. The latter is confusing..

Well, the reviewer may disagree, but we have throughout the paper argued not only maps but also the territory is a living structure, simply because they both have far more small things (or centres) than large ones. Both maps and the territory are a living structure; otherwise, maps would not be a living structure. In the following, let us highlight some of the related wordings.

In the Introduction, the very first sentence says “… (1) a map has a similar structure to the territory…” which implies that the territory is a living structure. In the fifth paragraph, it says “Firstly, it is argued that not only the territory but also the maps are a living structure with the recurring notion of far more smalls than larges.”…. At the end of the fifth paragraph, it says “…maps are largely about truth of the underlying living structure of the territory or the data.”

At the beginning of Section 4, “We contend in this paper that both maps of different scales and the territory are living structures with the inherent hierarchy of far more smalls than larges.”. At the beginning of Section 5, “Maps are essentially about the truth of the territory or that of the Earth’s surface, or more specifically the truth of the underlying living structure.”

Reviewer 2 Report

Comments and suggestions in attachment

Author Response

See the attached file namely RevisionDetails2B

Round 3

Reviewer 1 Report

I'm afraid there is a misunderstanding. The authors answered the other reviewer, but not my latest comments. The present revised version of the manuscript did not address my questions. Please send me a revised version that addresses my 2nd review, with a letter of explanation of changes. 

Reviewer 2 Report

I have already raised my considerations, in which the authors have replied -although not always convincingly.

I propose acceptance after minor corrections provided it is an opinion paper, which brings some fresh air in thinking about maps, but not a research nor a review paper.

Minor comments:

The link http://lifegis.hig.se/Sweden/ is missing some guide; also, it is hard to understand the differences from a -say- Google Map. And what is the difference between the two sides?

Still, the green diagonal line is not explained in Figure 2; please, write that the green line divides space into two sections, as the blue lines divide space into seven sections, etc.

Although in the title you claim that 'A Map Is a Living Structure ...', in the Conclusions you claim '...we have attempted to argue that not only maps but also the territory is a living structure. The latter is confusing.

Author Response

See the attached revision details report.

Round 4

Reviewer 1 Report

I accept the revised version.